# Overexpression of the Selective Autophagy Cargo Receptor NBR1 Modifies Plant Response to Sulfur Deficit

**DOI:** 10.3390/cells9030669

**Published:** 2020-03-10

**Authors:** Leszek Tarnowski, Milagros Collados Rodriguez, Jerzy Brzywczy, Dominik Cysewski, Anna Wawrzynska, Agnieszka Sirko

**Affiliations:** 1Department of Plant Biochemistry, Institute of Biochemistry and Biophysics Polish Academy of Sciences, Pawińskiego 5A St, 02-106 Warsaw, Poland; ljtarnowski@ibb.waw.pl (L.T.); Milagros.ColladosRodriguez@glasgow.ac.uk (M.C.R.); jurek@ibb.waw.pl (J.B.); blaszczyk@ibb.waw.pl (A.W.); 2Laboratory of Mass Spectrometry, Institute of Biochemistry and Biophysics Polish Academy of Sciences, Pawińskiego 5A St, 02-106 Warsaw, Poland; dominikcysewski@gmail.com

**Keywords:** autophagy, protein-protein interaction, sulfur deficit, transcriptome, *Arabidopsis thaliana*

## Abstract

Plants exposed to sulfur deficit elevate the transcription of *NBR1* what might reflect an increased demand for NBR1 in such conditions. Therefore, we investigated the role of this selective autophagy cargo receptor in plant response to sulfur deficit (-S). Transcriptome analysis of the wild type and NBR1 overexpressing plants pointed out differences in gene expression in response to -S. Our attention focused particularly on the genes upregulated by -S in roots of both lines because of significant overrepresentation of cytoplasmic ribosomal gene family. Moreover, we noticed overrepresentation of the same family in the set of proteins co-purifying with NBR1 in -S. One of these ribosomal proteins, RPS6 was chosen for verification of its direct interaction with NBR1 and proven to bind outside the NBR1 ubiquitin binding domains. The biological significance of this novel interaction and the postulated role of NBR1 in ribosomes remodeling in response to starvation remain to be further investigated. Interestingly, NBR1 overexpressing seedlings have significantly shorter roots than wild type when grown in nutrient deficient conditions in the presence of TOR kinase inhibitors. This phenotype probably results from excessive autophagy induction by the additive effect of NBR1 overexpression, starvation, and TOR inhibition.

## 1. Introduction

Autophagy is important for plant development, but also plays a role in the response to various environmental stresses. The autophagy process is tightly controlled by the cellular regulatory factors that respond to environmental stimuli (reviewed by [1]). One of the best characterized regulators of autophagy is the evolutionary conserved Target of Rapamycin (TOR) kinase that is on one hand activated by glucose and involved in the reciprocal regulation with plant hormones, and on the other hand, it stimulates such processes as translation, ribosome biogenesis, and cell proliferation (reviewed by [2,3]). In agreement with its overall growth promoting role, TOR kinase negatively regulates autophagy [4].

It is now widely accepted that autophagy is a highly selective process targeting specific cargo (proteins, organelles) that must be removed at the particular growth condition or life stage. The selectivity of autophagy is ensured by the involvement of the selective autophagy cargo receptors recognizing and targeting the load (the cargo tagged for degradation) to a double membrane vesicle called autophagosome. The subsequent fusion of the outer membrane of autophagosome with the tonoplast leads to cargo release into the vacuole and its degradation [5]. The best characterized cargo receptors in plants, NBR1, are evolutionary conserved. While animals have typically two members of the family, NBR1 and p62/SQSTM1, in plants there is a single member that is a structural hybrid of their two animal counterparts [6,7]. The plant NBR1 has several characteristic protein domains, PB1 involved in multimerization, ZF-ZZ-2, NBR1/FW (both of unknown function) and UBA domain involved in ubiquitin binding. The *C*-terminal UBA domain is doubled and it is unclear if this feature is critical for the function of NBR1. The protein has also the short amino acid motifs (LIR) that are responsible for binding to ATG8, which is necessary to dock the NBR1 to autophagosomes. Recent works with the plant NBR1-like proteins focused on their role in plant immunity and their interaction with plant pathogens [8,9,10]; however, their role in other aspects of plant growth and physiology had been neglected.

During the last decades many research groups have been studying various aspects of plant response to sulfur deficit, and several reports on metabolome and transcriptome changes in plants starved for sulfur are available [11,12]. These experiments led not only to a demonstration of the linkage between the S, P and N nutrition, but they were also sufficient to identify the regulatory circuits involved in plant response to S nutrition status. However, many elements of these circuits are unclear and not all factors involved in plants’ response to sulfur deficit have been identified, yet. Recently, it was discovered that sulfur starvation blocked the activity of TOR kinase in *Arabidopsis thaliana* via a not well-characterized downregulation of glucose metabolism [13]. The reduced TOR activity caused downregulated translation, lowered meristematic activity, and elevated autophagy. Furthermore, research from the same group pointed out the links between sulfate and cysteine availability and the abscisic acid (ABA) transduction pathway, especially in stomata closure. It was concluded that the positive effect of sulfate or cysteine on stomatal closure was mediated by ABA because of the sulfur requirement for ABA synthesis [14,15,16]. Previously, we reported that expression of the gene encoding the selective autophagy cargo receptor Joka2 (NtNBR1) in tobacco was induced in plants exposed to sulfur deficit [7]. The links between plant NBR1 and sulfur availability were additionally suggested by the fact that Joka2 was identified as a partner of the UP9C protein encoded by a gene strongly induced by sulfur starvation [17,18]. The UP9C protein is a member of the plant-specific family of LSU (Response to Low SUlfur) –like proteins, identified as important stress hubs involved in multiple protein-protein interactions [19,20].

Experimental data for the existence of links between plant NBR1 and nutrient deficiency and particularly its role in sulfur deficient conditions are rather scarce. Therefore, we decided to investigate the consequences of constitutive ectopic overexpression of NBR1 in Arabidopsis in sulfur deficient conditions and to identify the proteins co-purifying in complexes with NBR1 in the shoots and roots of plants grown in sulfur sufficient and sulfur deficient conditions.

## 2. Materials and Methods

### 2.1. Plant Material

Columbia (Col-0) *Arabidopsis thaliana* was used as the parental line for the NBR1 overexpressing lines (NBR1-OX) and *nbr1* deletion lines (*nbr1*-KO). Expression cassettes for OX7.5 and OX2.3.5 lines containing the entire open reading frame of *NBR1* and for M6.2 line containing residues 1-614 (C-terminally truncated NBR1 lacking the UBA domains) fused to TAP tag at *C*-terminus were generated using the binary vector pC-TAPa (pYL436) [21]. The TAP line, used as a control in TAP-MS experiment, was generated using the empty pN-TAPa vector [21]. Expression cassette for NBR1-YFP line construction was generated using the binary vector pH7YWG2 [22]. In the above lines, the transgenic *NBR1* is under the control of the constitutive promoter (35S from CaMV). Deletions in *NBR1* gene were generated using the CRISPR/Cas9 method [23]. The genomic regions covering the deletions were amplified and sequenced, and the *nbr1*-KO lines will be described in a separate paper. Briefly, several independent deletions (e.g., KO1, KO3) starting about 140–170 nt upstream translation initiation codon and covering the region of the first 171–205 residues (depending on the line) were obtained (Tarnowski et al. submitted [24]). The *Nicotiana tabacum* J4-1 and J5-3 transgenic lines constitutively overexpressing Joka2 (NtNBR1) in the LABarley 21 background were described previously [7].

### 2.2. Plant Growth Conditions

The media composition is provided in Appendix A. Plants were usually grown in hydroponic conditions in the controlled conditions of a plant growth chamber (22 °C 8h day/18 °C 16h night). Seedlings grown hydroponically (in 0.5× Hoagland-based media) were gently agitated (22 °C 12 h day/12 h night) in 24-Well Suspension Culture Plates Cellstar^®^ (Greiner Bio-One, Frickenhausen, Germany). To compare the roots growth, the media were supplemented with 15 μM glucose; when indicated, they also contained TOR inhibitors: rapamycin (10 μM), Torin1 (1 μM) or AZD8055 (1 µM). The dose of inhibitors was adapted from different literature data and preliminary verified. For instance, from three tested concentrations of AZD8055 (0.5 µM, 1 µM and 1.5 µM) the difference in growth between the lines was observed only in 1 µM. The lower concentration did not have effect on root length, while the highest inhibited the root length of all lines. The plants for microarray and TAP-MS experiments were grown in Araponic boxes in 0.5 AB-based media for four weeks (with weekly changes of the nutrient sufficient medium [nS]) before transferring to the [nS] and [–S] for either 4 or 10 additional days.

### 2.3. Transcriptome Analysis

RNA preparation (separately from shoots and roots of plants grown in –S for four days or in sufficient S), cDNA synthesis, microarray experiment, and data analysis were performed as described previously [24]. Three independent biological replicates of shoots and roots were analyzed using the Arabidopsis Gene 1.1 ST Array Strips (Affymetrix, Santa Clara, CA, USA) at the Laboratory of Microarray Analysis, the Institute of Biochemistry and Biophysics Polish Academy of Sciences. Lists of differentially expressed genes (DEGs) between biological variants, with fold changes of at least ± 1.5, and *p*-values < 0.05, were created after applying a 3-way ANOVA to the data. The data discussed in this publication are accessible through GEO Series accession number GSE122705.

### 2.4. Reverse Transcription-Real Time Quantitative Polymerase Chain Reaction (RT-qPCR)

qPCR was performed using cDNA isolated from the material used for transcriptomic analysis and the following pair of oligonucleotides (5′-TCAGGTGTACTCGCCCTAAA-3′) and (5′- GCGAACCACTGTTCCTCATT-3′) as the forward and reverse primers, respectively, in a PikoRealTM Real-Time PCR System (Thermo Fisher Scientific, Waltham, MA, USA) with Luminaris Color HiGreen qPCR Master Mix (Thermo Fisher Scientific). The primers are complementary to the coding region of *NBR1*. *Actin 2* (At3g18780) was selected as a constitutively expressed gene to normalize the quantity of total RNA present in each sample. Relative gene expression levels were calculated using the delta-delta Ct method [25]. The qPCR was carried in 3 biological replicates (each in 3 technical replicates to assess operator variance).

### 2.5. Western Blot Analysis

Plant material was homogenized in 100 μL of the extraction buffer (1M Tris-HCl, pH 8.0) with 1 μL of Protein Inhibitor Cocktail (Merck) and centrifuged (13 krpm, 4 °C, 10 min). Proteins from the supernatant were separated on 8% SDS-PAGE gels, transferred to the nitrocellulose membrane (BioRad, Hercules, CA, USA), and visualized by staining with Ponceau S (loading control). Membranes were blocked with 5% non-fat dry milk and then probed with rabbit anti-AtNBR1 polyclonal IgG, custom generated against the 165-aa *N*-terminal fragment of NBR1 (GenScript, Carlsbad, CA, USA). Goat anti-rabbit IgG (Sigma-Aldirch, St. Louis, MI, USA) coupled to the alkaline phosphatase were used as secondary antibody. Protein bands were visualized by adding 5 mL BCIP/NBT Solution (Bioshop, Burlington, Ontario, Canada) at room temperature.

### 2.6. Confocal Microscopy

A Nikon Eclipse TE2000-E inverted confocal microscope was used for all observations. Acidic speckles (such as autophagosomes) were stained for 15 min with 1 μg/mL acridine orange (Invitrogen, Carlsbad, CA, USA) and observed using a 543 nm laser (helium-neon laser; Melles Griot, Cardiff, United Kingdom) and a 605/75 filter. Stomata were observed in visual light. Fluorescence from NBR1-YFP was observed in fresh leaves using a 488 nm laser (Coherent sapphire 488-20 CDRH) and 515/30 filter. Bimolecular fluorescence complementation (BiFC), used to test protein-protein interactions in planta, was monitored 2 days after agroinfiltration of Nicotiana benthamiana leaves with Agrobacterium tumefaciens cells (strain GV3101) transformed with the tested combination of plasmids. The plasmids encoded the *N*-terminal (pSITENYFPC1) or *C*-terminal (pSITECYFPC1) part of YFP [25] linked to the proteins or protein fragments being investigated, and 35S::H2B-RFP to stain the nuclei, if indicated. Bacteria were grown for 24 h at 28 °C in YEB medium supplemented with 10 μg/mL rifampicin and 50 μg/mL spectinomycin prior to agroinfiltration. Interactions were tested using a 488 nm laser (Coherent sapphire 488-20 CDRH) and a 515/30 filter. For red fluorescent protein (RFP) a 543 nm laser (helium-neon laser; Melles Griot) and a 605/75 filter were used. The image data were analyzed with an EZ-C1 3.90 FreeViewer (Nikon Corporation, Tokyo, Japan) and ImageJ (version 1.41, National Institutes of Health, Bethesda, MA, USA). Plasmids used in the experiments are listed in Appendix A.

### 2.7. Yeast-Two-Hybrid (Y2H) Analysis

Y2H analysis was performed as previously described [17]. Plasmids used in the experiments were based on pDEST22 and pDEST32 (Thermo Fisher Scientific) and are listed in Appendix A.

### 2.8. TAP-MS Experiment

Sample preparation for MS: The NBR1-TAP (line for NBR1 complexes isolation) and TAP (control line) were grown in -S for either four days (4d-S) or for 10 days (10d-S) or in sufficient S (nS). Each portion of plant material was prepared in biological duplicates and analyzed in duplicates. Total protein was isolated separately from shoots and roots of two-four plants each and purified according to the procedure described earlier [21]. Samples were precipitated with −20 °C cold acetone.

LC MS/MS analysis: Precipitated proteins were dissolved in 50 μL of 100 mM ammonium bicarbonate buffer, reduced with 0.5 M (5 mM final concentration) tris(2-carboxyethyl)phosphine for 1h at RT, blocked with 200 mM S-Methyl methanethiosulfonate (10 mM final concentration) for 10 min at RT, and digested overnight with 10 ng/mL trypsin (CAT NO V5280, Promega) at 37 °C. Finally, to stop digestion, trifluoroacetic acid was added at a final concentration of 0.1%. The mixture was centrifuged at 4 °C, 14,000*g* for 30 min, to remove the remaining solid. MS analysis was performed by LC-MS in the Laboratory of Mass Spectrometry (IBB PAS, Warsaw, Poland) using a nanoAcquity UPLC system (Waters, Milford, MA, USA) coupled to an Orbitrap Velos LTQ mass spectrometer (Thermo Fisher Scientific, Waltham, MA, USA). The mass spectrometer was operated in the data-dependent MS2 mode, and data were acquired in the m/z range of 300–2000. Peptides were separated by a 180 min linear gradient of a 95% solution A (0.1% formic acid in water) to a 35% solution B (acetonitrile and 0.1% formic acid). The measurement of each sample was preceded by three washing runs to avoid cross-contamination. The final MS washing run was searched for the presence of cross-contamination between samples. Data were searched with the Max-Quant (Version 1.5.7.4, Max-Planck-Institute for Biochemistry, Martinsried, Germany) platform search parameters: match between runs (match time window 0.7 min, alignment time 20 min), enzyme: trypsin/p; specific; max missed 2, minimal peptide length 7-aa, variable modification: methionine oxidation, N-term acetylation, fixed: cysteine alkylation, main search peptide tolerance 4.5 ppm, protein FDR 0.01. The reference *Arabidopsis thaliana* proteome database from TAIR was used (downloaded on 2017.03.17, 35 386 entries). The mass spectrometry proteomics data have been deposited to the ProteomeXchange Consortium and are available via ProteomeXchange with identifier PXD016026. Semiquantitative analysis of TAP results used protein intensity comparison [26]. Protein abundance was defined as the mean signal intensity calculated by MaxQuant software for a protein (mean value from two or one replicates) divided by its molecular weight. Specificity was defined as the ratio of protein signal intensity measured in the bait purification to the background level (which is the protein signal intensity in the negative control purification; the background level was arbitrarily set to 1 for proteins not detected in the negative control).

### 2.9. Bioinformatic Tools and Statistical Analysis

Venn diagrams were drawn using the online web tool (http://bioinformatics.psb.ugent.be/webtools/Venn/). GO Term enrichments were calculated on the agriGO v2.0 server (China Agricultural University, Beijing, China) [27]. Gene annotations were downloaded from The Arabidopsis Information Resource (TAIR) and Frequencies of Functional Categorization were calculated using TAIR’s web tool (https://www.arabidopsis.org/tools/bulk/go/index.jsp). Interaction networks were analyzed with the Cytoscape software (version 3.7.1, Institute for Systems Biology, Seattle, WA, USA) [28] using the BioGRID database version 149 or newer [29]. The statistical significance of the RT-qPCR results and quantification of AO-stained red spots were tested by one-way ANOVA and Fisher’s LSD post-hoc test using Statistica 12 software ((StatSoft Polska, Krakow, Poland). Sets of genes and proteins were analyzed online with VirtualPlant 1.3 software platform (http://virtualplant.bio.nyu.edu) [30].

## 3. Results

### 3.1. Sulfur Deficit Increases the Amount of NBR1 mRNA

We have previously reported that sulfur deficit increased the amount of *Joka2* (*NtNBR1*) transcript encoding selective autophagy cargo receptor NBR1 in tobacco [7]. A similar increase of *NBR1* transcript during sulfur deficit was observed in shoots and roots of *Arabidopsis thaliana* plants grown in the conditions of our experiment (Figure 1). The amount of the *NBR1* transcripts (corresponding to both, the intrinsic and ectopically expressed gene under constitutive promoter) was also increased by -S in the lines overexpressing *NBR1.* We have detected NBR1 proteins (intrinsic and ectopically expressed) in shoots and roots of plants from both growth conditions, (Appendix A).

### 3.2. Seedlings From –S Have More Acridine Orange-Stained Red Spots

Acridine Orange (AO) is a membrane-penetrable fluorescent dye useful for visualization of acidic compartments [7]. In the AO stained cells, the cytoplasm and nucleus emit green fluorescence, while the acidic compartments fluoresce in red. AO is not an ideal marker for autophagy but it has already been used for such purposes, especially after the comparison of the ratio of red/green fluorescence [31,32]. In this study, AO has been used to visualize acidic spots in the seedlings of several Arabidopsis lines grown in nS and -S conditions. The seven-day-old seedlings grown in -S had an increased amount of red staining in comparison to the seven-day-old seedlings from the same line grown in nS (Figure 2A). In both conditions, the NBR1-overexpressing seedlings had more AO-stained red spots than the WT seedlings grown in the corresponding conditions. Moreover, the AO staining of NBR1-YFP seedlings, producing the NBR1-YFP fusion protein, indicated that about 30% of the green spots overlapped with the red speckles (Figure 2B). This suggests that a significant portion of the NBR1-YFP did indeed co-localize with the acidic compartments, presumably autophagosomes.

### 3.3. Response to –S in WT and OX7.5 at the Level of Transcription

We have identified 4318 and 2794 differentially expressed genes due to –S conditions (-S DEGs) in shoots of WT and OX7.5, respectively, and of 4669 and 4602 –S DEGs in roots of WT and OX7.5, respectively (Appendix A). For each part of plants, we obtained the following sets of –S DEGs: upregulated in both lines, upregulated only in WT, upregulated only in OX7.5, downregulated in both lines, downregulated only in WT and downregulated only in OX7.5 (Appendix A). In shoots, the common –S DEGs represent 42% and 34% of all genes up- or down-regulated, respectively, while in roots they represent 50% and 41% of all genes up- or down-regulated, respectively. Numerous genes were differentially expressed in response to -S in only one of the two tested lines (Figure 3A). Analysis of the overrepresented terms in 12 sets of –S DEGs indicated that cytoplasmic ribosome protein gene family is overrepresented in all sets of –S DEGs upregulated in roots (Table 1). Of 208 genes from the cytoplasmic ribosome protein gene family (number of genes according to [33]) there were 20, 43 genes upregulated in single lines (WT and OX7.5, respectively), while 71 genes were upregulated in both lines (Figure 3BC; Appendix A). The upregulated genes are scattered throughout both ribosomal subunits (Figure 3C).

In order to get insight into the possible mechanisms responsible for the transcriptional differences in response to –S between the WT and OX7.5 lines, we analyzed the interaction networks of –S DEGs in WT and OX7.5. For each node in the Biogrid database, we calculated the number of its total interactions (degree.total), number of its interactions with –S DEGs in WT (degree.-S) and number of its interactions with -S DEGs in OX7.5 (degree.-S in OX). When number of -S DEGs was plotted against number of total interactions RPT2a (At4g29040) and UBQ3 (At5g03240) were identified as the most significant hubs in both interaction networks analyzed (Figure 4A). However, for several minor hubs we observed differences between WT and NBR1-OX in the number of interactions with –S DEGs. We identified 5 and 11 hubs with more and fewer interactions, respectively, in the NBR1-OX network than in WT network (Figure 4B). Among the 5 hubs enriched in interactions in the OX7.5 network, 4 are annotated as LRR (Leucine-Reach Repeat) protein kinases. While among 11 hubs with a reduced number of interactions in the NBR1-OX network, 7 are annotated as transcription factors belonging to the TCP (TEOSINTE BRANCHED 1, cycloidea and PCF transcription factor) family (Appendix A). These results suggest that excess of NBR1 modulates plant response to –S by indirect or direct control of LRR kinases and/or TCP transcription factors. To our knowledge, none of these elements have been previously associated with –S response.

The difference in –S response between WT and OX7.5 lines can be partially explained by the fact that many genes regulated by –S might be regulated by NBR1 overexpression. Indeed, a significant common part was revealed between these two sets of DEGs (Figure 5, Appendix A). In shoots, the number of overlapping genes was 208 (70% of OX DEGs) and 115 (53% of OX DEGs) for up- and downregulated, respectively. While in roots, it was 160 up- and 214 downregulated genes, corresponding to about 38% and 42% of the respective OX DEGs in roots.

### 3.4. Proteins co-Purifying with NBR1-TAP

The next goal was to identify the proteins co-purifying with NBR1 in shoots and roots in different growth conditions using the TAP-MS experiment. The lists of identified proteins are included in Appendix A. The Venn diagrams comparing the roots and shoots interactomes, each in three conditions (nS, 4d-S, 10d-S), are shown in Figure 6. The set of condition-specific proteins is apparently larger in roots than in shoots. In shoots, for 242 proteins detected totally in all three growth conditions, 140 proteins (almost 58%) were detected in all of them; while in roots, there were only 41 such proteins (23%) for 182 total proteins. Besides, the number of proteins identified only in roots was smaller in each condition than the number of proteins identified only in shoots. The ribosomal proteins co-purifying with NBR1 are marked in Figure 3C and listed in Appendix A.

### 3.5. Interactions Network of Proteins Co-Purifying with NBR1-TAP

In order to further characterize the NBR1 interactome, we analyzed known interactions of all proteins co-purifying with NBR1 from any plant part (root or shoot) or growth condition (nS or -S). Among 136 proteins from such a combined set, 115 have known interactions in the BioGRID database ver. 174. For all proteins co-purifying with NBR1 and their first neighbors, we counted the number of interacting proteins that co-purify with NBR1 and the total number of interactors in the BioGRID database. Plotting these values allowed for the selection of the network hubs having a high number of interactors among the NBR1 co-purifying proteins compared to the total number of their interactors (Figure 7). The eight top proteomic hubs (Hub1-Hub8) are sorted in Table 2 according to a decreasing number of common interactors with NBR1; while their targets identified in the TAP-MS experiment with NBR1 are listed in Appendix A. Ubiquitin (UBQ3, At5g03240) is a hub number 1 having known interactions with 59 NBR1 co-purifying proteins. Hub2 (nucleoporin NUP43; At4g30840) and hub3 (WD40 protein RAE1; At1g80670) are close to the top of the list and share the high number of interacting proteins with the NBR1 co-purifying proteins (24 and 20, respectively). Hub4 (proteasomal subunit RPT2a; At4g29040) and hub5 (general regulatory factor from the 14-3-3 family GRF1; At4g09000) have known interactions with 16 and 12 NBR1 co-purifying proteins, respectively. RPT2a was identified also as a “transcriptomic” hub of –S DEGs (see above). Each of the next two hubs, hub6 (clathrin adaptor protein AP2M, At5g46630) and hub7 (histone H3 acetyltransferase IDM1/ROS4, At3g14980), share 11 interactors with the NBR1 co-purifying proteins NBR1; while hub8 (human protein RAD23A) was previously reported to interact with 10 proteins from our TAP-MS experiment.

### 3.6. Verification of NBR1 Interaction with RPS6

The ribosomal protein S6 was present in the sets of the previously reported interactors of proteomic Hub3 (RPT2a) and Hub5 (AP2M). Besides, it is linked with autophagy signaling via the TOR-RPS6 kinase-RPS6 pathway. Therefore, it was selected for confirmation of its direct binding to NBR1. We decided to verify the interaction of NBR1 with RPS6 using two strongly conserved isoforms of this protein, RPS6A (not detected in TAP-MS experiment) and RPS6B. Indeed, the BiFC experiment indicated that NBR1 interacts with both of them (Figure 8). In NBR1, the region involved in the interaction with RPS6 was narrowed down to the 314 amino acid fragments from amino acid G92 to R405, encompassing the ZZ domain along with both flanking interdomain regions. In RPS6, the area of interaction was limited to a fragment of 87 amino acids between S109 and R195 (identical in both isoforms of RPS6). The NBR1-RPS6 interaction was additionally verified in the yeast two-hybrid (Y2H) experiments. These results indicated that the interaction of NBR1 with the RPS6 isoforms was not mediated by the UBA domain, and it was probably not the classical target- autophagy receptor interaction. Interestingly, the interaction between RPS6A and NBR1 was observed in both the cytoplasm and nucleus, while RPS6B-NBR1 interactions were observed only in the cytoplasm (Figure 8D).

### 3.7. NBR1-OX Lines are Oversensitive to TOR Inhibitors During Nutrient Starvation

Interestingly, the NBR1 overexpressors have significantly shorter roots than WT when grown in -S or in nitrogen deficient (-N) conditions in the presence of TOR kinase inhibitors (Figure 9A, Appendix A). A similar phenomenon was observed also in *Nicotiana tabacum* J4-1 and J5-3 transgenic lines overexpressing a tobacco homolog of NBR1 (NtNBR1-OX), indicating that the mechanisms responsible for the observed phenotype are common for both plant species (Figure 9B). Staining of the Arabidopsis seedlings grown in –S in the absence and presence of Torin1 (one of the TOR inhibitors) indicated a strong increase of red AO-stained spots in the plantlets treated by Torin1 (Appendix A).

## 4. Discussion

### 4.1. NBR1 Expression and Autophagy Flux in NBR1-OX Lines and –S Conditions

In the NBR1-OX lines used in this work, the ectopic copy of NBR1 is under control of the double 35S promoter of the cauliflower mosaic virus, however differences in the NBR1 transcript levels were observed between OX7.5 and OX2.3.5. Only OX7.5 had more NBR1 transcripts than the WT line in shoots and roots. In the conditions of our experiment the sulfur starvation stress (4 days in –S condition) upregulated the amounts of NBR1 transcripts in shoots and roots of all lines. The reason for the increased amounts of *NBR1*-containing transcripts in –S is not clear. It might result either from increased transcription of the intrinsic *NBR1,* as well as from the increased stability of the transcript.

As expected, NBR1-OX lines had not only intrinsic NBR1 but also transgenic NBR1-TAP protein (Appendix A). Both proteins appeared larger than expected. An increase of the size of NBR1 from *Arabidopsis thaliana* has been also recently reported [35]. The plausible explanation of the observed discrepancy would be posttranslational modification; however, it remains to be determined. Interestingly, the seedlings of both OX lines had more autophagosomes (red spots stained by AO) than WT, and the number of autophagosomes was increased by –S in each line (Figure 2A). This increase was insignificant in OX2.3.5, significant in WT and very strong in OX7.5. Since NBR1 itself is an autophagy substrate [6,35] one could expect a smaller amount of this protein when the number of autophagosomes is increased and the autophagy flux is induced. The apparent discrepancy suggests that an excess of NBR1 accumulates not only in autophagosomes but also in non-acidic structures, which are not accessible to degradation. Only a partial overlap of the green and red signals in NBR1-YFP overexpressors stained with AO (Figure 2B) additionally supports this conclusion. Therefore, the level of NBR1 in lines constitutively overproducing this protein might not be an accurate indicator of the autophagy flux.

### 4.2. Transcriptional Response to –S in WT and OX7.5

Our observation that -S induces NBR1 expression prompted us to investigate the role of NBR1 in such conditions. However, interaction network analyses of the two sets of –S DEGs (one from the WT and one from NBR1-OX) revealed their similarity and led to the identification of only two hubs significantly enriched in interactions with –S DEGs regardless of the line, RPT2a and UBQ3 (Figure 4A). RPT2a is an AAA type ATPase of the 26S proteasome subunit which, among other functions, interacts with transit peptides of chloroplast proteins [36]. Identification of these hubs is not surprising because proteasomal degradation might control the activity of SLIM1, a transcriptional factor adjusting plant response to sulfur deficit [37]. The part of –S response controlled by proteasome is rather independent of NBR1, and it seems that NBR1 only fine-tunes the plant response to –S. There might be also other not yet recognized connections between UPS and plant response to –S.

Despite the overall similarity of the networks of –S DEGs in WT and OX7.5, differences in the enrichment of some hubs in both networks were observed (Figure 3B, Appendix A). Such differences might provide an additional clue for the function of NBR1. The first set of these hubs encompasses 4 LRR kinases (At5g45780, At5g16590, At2g45349, At3g23939), which are more enriched in interactions among –S DEGs in OX7.5 than in WT. The family of LRR kinases in Arabidopsis consists of about 200 members, which are involved in sensing various signals and play a role in plant growth and development (for review [38]). The LRR kinases are classified according to the size of their extracellular domain into seven categories grouped in two types, ligand-recognizing receptors and co-receptors, which form heterodimers playing an important role in fine-tuning the signaling by the receptor [39]. Heterodimers’ pairs preferences are unknown for most of the LLR kinases, including the co-receptor kinases identified as differentiating hubs for –S DEGs from WT and OX7.5. An elevated transcription of genes, whose products are in the interactomes of the mentioned above LRR kinases, might indicate that the kinases themselves are more activated in OX7.5 exposed to –S than in WT. It is unclear at what level the NBR1 protein might influence the activity of LRR kinases and such hypothetical links should be further investigated.

The second set of hubs is more enriched in interactions with –S DEGs in WT than in OX7.5. This set of hubs is more divergent, but it contains several transcription factors from the TCP family known to closely interact with members of plant hormone signaling pathways (for review of the TCP family see [40]). Among 11 hubs associated with the change of –S response in NBR1-OX (reduced number of interaction) as compared to the –S response in the WT, 7 belong to the TCP family, TCP14 (At3g47620), TCP13/PTF1 (At3g02150), TCP4 (At3g15130), TCP15 (At1g69690), TCP2 (At4g18390), TCP10 (At2g31070), and TCP23 (At1g35560). The TCP family consists of 24 members in Arabidopsis and is divided into two groups, both playing a complex role in plant development and immune defense [41,42]. TCP14 is the most targeted host protein by the effectors of many plant pathogens [43]. The reduced number of –S DEGs from OX7.5 having reported interactions with these proteins might suggest that these transcription factors are not activated by –S to the same extent in OX7.5 compared to WT. The involvement of LRR receptor kinases or TCP in –S response has not yet been reported, however the overlap between plant responses to a variety of stresses is a known phenomenon [44].

This study indicated that sulfur starvation results in changes in expression of the genes encoding ribosomal proteins. Many genes encoding cytosolic ribosomal proteins were induced in roots, while many genes for nuclear encoded plastid genes were downregulated in shoots. NBR1 overexpression changes the pattern of expression of many genes from the first group. Interestingly, the difference in expression of ribosomal proteins was previously observed in plants starved for phosphorus and iron [45]. To our knowledge, no published data linked so far selective autophagy and NBR1 with ribosome remodeling.

### 4.3. Proteins Co-Purifying with NBR1-TAP

The interaction network analysis indicated that the proteins co purifying with NBR1-TAP are not randomly distributed around the Arabidopsis interactome but seem to interact with specific proteins. Eight hubs having at least 10 known interactions with proteins identified in our TAP-MS experiment were identified (Table 1). Most of them are involved in RNA export from the nucleus or in proteolysis. The highest number of interactors among the NBR1 co-purifying proteins has ubiquitin (represented by UBQ3, At5g03240), designated here as hub1. Furthermore, the number of proteins co-purifying with NBR1 that were previously identified as a part of the UBQ3 interactome highly increases in roots after prolonged sulfur starvation, suggesting an increase in protein ubiquitination in –S conditions. An increase of protein ubiquitination in plants grown in -S is quite plausible (Wawrzynska and Sirko, unpublished data). Hub2 and hub3 have WD40 domains and are part of the nuclear pore complex (NPC). Interestingly, Hub 3 (RAE1; At1g80670) is a component of a nuclear pore [46], but it is also involved in proteolytic protein degradation controlling the level of ABA receptor RCAR1 and negative regulation of ABA signaling [47]. Hub4 (RPT2a, At4g29040) is AAA type ATPase of the 26S proteasome subunit which, among its other roles, interacts with transit peptides of chloroplast proteins [35]; while hub 5 (GRF1, At4g09000) is a member of a large family of GRFs (general regulatory factors, 14-3-3 family), which are involved in protein-protein interactions serving as adapters, activators, and repressors in diverse cellular processes (for recent reviews see [48,49,50]). Hub 6 is the adaptor protein 2 (AP2M, At5g46630) which is composed of clathrin binding domains and apparently involved in endocytosis [51]. AP2M is missing any ubiquitin binding domains (INTERPRO domains listed by TAIR). Therefore, AP2M probably recognizes its targets by a different mechanism than via ubiquitin binding. Thus, it seems possible that some common interactors of AP2M and NBR1 may be bound by NBR1 in an UBA independent mode as well. Checking this possibility seems to be an interesting goal for future research. Hub7 (IDM1/ROS4; At3g14980) is a histone H3 acetyltransferase negatively regulating DNA methylation that might reduce gene silencing. The links of this hub with selective autophagy remain unclear. Finally, it may seem surprising that the human protein RAD23A appears as a proteomic hub in Arabidopsis (Hub8). However, this protein was used for selection of ubiquitinated proteins in Arabidopsis and the list of its interactors was published [34]. Taking into account that NBR1 protein contains two ubiquitin binding UBA domains, similar to the human RAD23A protein, it seems likely that NBR1 and HsRAD23A may recognize ubiquitin moieties on the same target proteins. Besides, NBR1 itself was on the list of HsRAD23A interactors [34]. In summary, the presence of proteins involved in protein degradation among proteomic hubs may result from sharing a common feature with NBR1, namely, recognition of ubiquitin, with exception of hub 5 (vesicular transport/endocytosis) and hubs 1 and 2 (transport via nuclear pores).

### 4.4. NBR1 Might be Involved in the Regulation of Ribosome Composition in -S

Identification of the proteins targeted by NBR1 might help us to understand its role in –S conditions. Several proteins were co-purifying with NBR1 only in –S conditions (Figure 6, Appendix A). Among them there were ribosomal proteins, of which some were also upregulated in –S (Figure 3). One of these ribosomal proteins, RPS6, is known to be involved in TOR signaling and regulation of translation efficiency as a part of the TOR-S6K-RPS6 signaling pathway that is responsible for the adjustment of cellular translation to the metabolic needs sensed by TOR [52]. Moreover, the increased level of RPS6 protein found in *atg5* mutant of *Arabidopsis thaliana* suggests that this protein could be a subject of autophagic degradation [53]. Besides, the recent report that autophagy is required for recycling of ribosomal proteins RPS6, RPL37 in *Chlamydomonas reinhardtii* is in agreement with this assumption [54]. Therefore, RPS6 was selected for verification of its direct interaction with NBR1.We confirmed a direct RPS6-NBR1 binding in yeasts and in plants (Figure 8). This novel direct interactor of NBR1 binds to the NBR1 fragments spanning the Zn finger (ZZ) domain flanked by its interdomain regions. Both isoforms of RPS6 (RPS6A and RPS6B) can interact with NBR1. The lack of requirements for the UBA domain of NBR1 for binding suggests that RPS6 possibly does not need prior ubiquitination for interaction with NBR1 and that it might be another type of interplay than the classical “ubiquitinated target - autophagy receptor”. The region of RPS6 interacting with NBR1 is located in the central part of the protein, and it is apparently different from the C-terminal region, known to be phosphorylated by S6K [55]. Therefore, this interaction might not be dependent on the phosphorylation status of RPS6 controlled by S6K. It is commonly accepted that RPS6 regulates translational capacity. Therefore, precise control of RPS6 activity (or intracellular localization) might be important for metabolic reprograming needed for plant response to –S. Nevertheless, the biological significance of this novel interaction of NBR1 remains to be determined. Direct interaction of NBR1 with other ribosomal proteins requires further studies, however both, the transcriptomic and proteomic data suggest that NBR1 could be involved in the regulation of ribosome composition in –S (Figure 3). Multiple studies indicated ribosomes specialization (ribosomes of different composition or ribosomes lacking specific proteins) depending on the tissues or external conditions in different organisms [56].

### 4.5. NBR1-OX Plants Were Oversensitive to –S and to –N in the Presence of TOR Inhibitors

The TOR kinase is a well-known negative regulator of autophagy, as phosphorylation of autophagy-related protein 13 (ATG13) by TORC1 inhibits the formation of the autophagy initiation complex [57]. TOR is activated by nutrients and hormones, and it is repressed by starvation, energy deprivation, and stress [58,59]. TOR regulates many metabolic pathways including photosynthesis [60], the cell-cycle, cell-wall modifications, senescence, central energy metabolism, carbon and lipids metabolism and secondary metabolism [61]. It also induces the transcription of nuclear-encoded plastid ribosomal proteins [62]. Reduced expression of numerous such genes observed in –S conditions in the shoots of all lines (not shown) would indicate the reduction of TOR activity in –S conditions. Moreover, TOR positively regulates initiation of translation in plants, possibly by stimulation of phosphorylation of the translation initiation factor eIF3h via S6K [63]. The simplest explanation of the observed phenotype (oversensitivity of NBR1 overexpressors to TOR inhibitors in –S conditions) is an excessive induction of autophagy. This assumption is supported by the observation of autophagosomes (precisely, AO-stained red speckles) in the roots of such plants (Appendix A). The massive red staining suggests the presence of numerous autophagosomes due to a high induction of autophagy.

## 5. Conclusions

Concluding, our results suggest that NBR1 can fine-tune plant response to sulfur starvation. We propose that the modulation of the level (or of the activity) of different ribosomal proteins might be one of the processes controlled by NBR1 in such conditions. This conclusion is supported by the following observations: (i) partial overlap of gene expression changes due to NBR1 overexpression with the gene expression changes in the wild type plants exposed to sulfur deficit, (ii) the overrepresentation of ribosomal proteins in the set of NBR1 partners in –S conditions and (iii) direct NBR1-RPS6 binding. However, the effect of NBR1 on plant response to sulfur deficit is pleiotropic and this protein might be involved in selective degradation of multiple protein targets, not only the ribosomal proteins. The identities of most of these targets remain to be deciphered.

## Figures and Tables

**Figure 1 cells-09-00669-f001:**
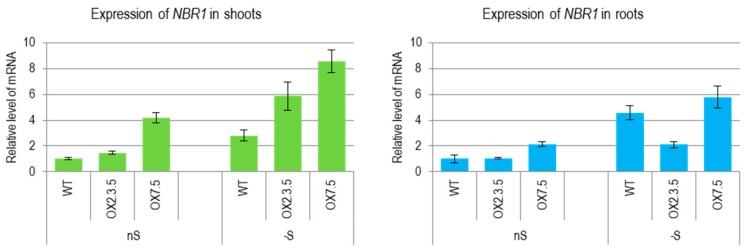
The expression level of NBR1 in wild type (WT) and NBR1-overexpressor lines (OX2.3.5, OX7.5) grown in sulfur sufficient (nS) and sulfur deficient (-S) conditions. Results of reverse transcription quantitative PCR (RT-qPCR) using the shoots and roots material normalized to WT grown in nS. The significantly different samples are indicated with different letters.

**Figure 2 cells-09-00669-f002:**
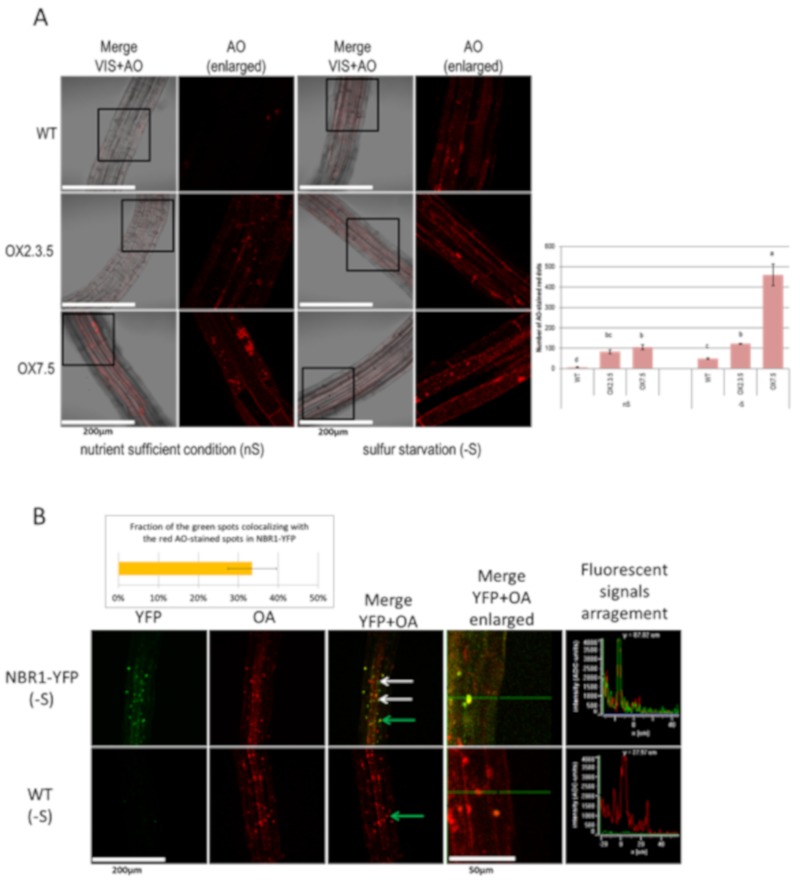
Comparison of the roots stained with acridine orange (AO). (**A**) Sulfur starvation increases the number of acidic compartments stained with AO in WT, OX2.3.5, and OX7.5. The graph showing the average number of AO-stained red spots calculated with ImageJ software from three independent experiments (three pictures per each line in each experiment; *n* = 9) is shown below the representative confocal fluorescence microscopy photographs. The significantly different samples are indicated with different letters (**B**) NBR1-YFP co-localizes with many acidic compartments stained with AO (examples are shown by arrows: the green arrow points to the dot for which fluorescence signals were arranged in graph view). The results showing the quantification of the colocalization of the green spots with the red spots are from three independent experiments (three pictures each; *n* = 9).

**Figure 3 cells-09-00669-f003:**
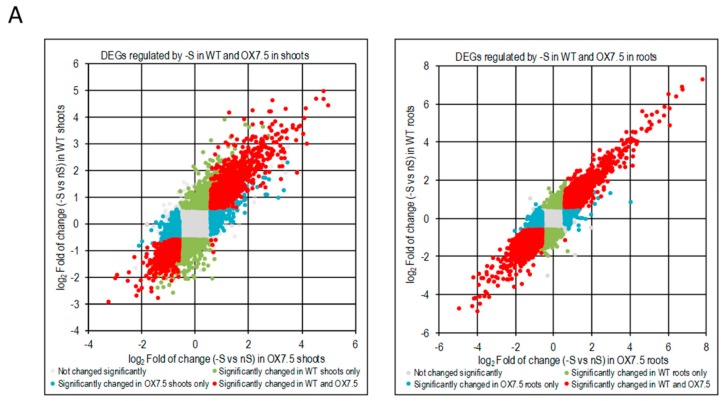
Genes regulated by sulfur starvation in shoots and roots of WT and OX7.5. (**A, B**) Scatterplots showing fold of change of individual genes in –S. DEGs are marked in colors, as indicated in the legends, while the genes not changed significantly are denoted in pale grey. (**B**). Scatterplot comparing regulation of expression of the cytosolic ribosomal protein gene family by –S in the roots of wild type and NBR1-OX line. (**C**) The KEGG chart of *Arabidopsis thaliana* ribosomes. The proteins found in TAP-MS experiment as co-purifying with NBR1 in nS (yellow) or in -S only (orange). Changes of gene expression of the paralogues of ribosomal genes in –S are shown below the respected proteins using the colors indicated in the legend of the chart in Panel B (green, upregulated only in WT; blue, upregulated only in OX7.5; red, upregulated in both lines).

**Figure 4 cells-09-00669-f004:**
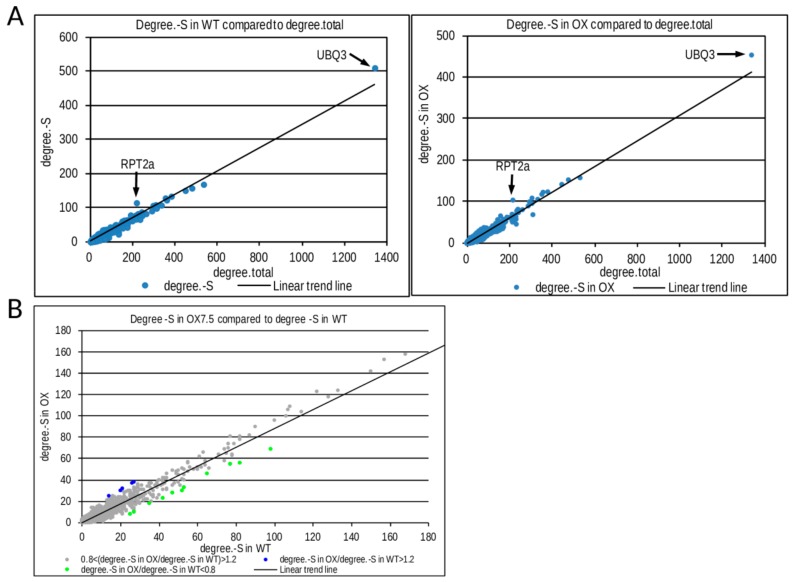
Results of interaction network analysis of genes regulated by sulfur starvation in WT and OX7.5. (**A**) Identification of RPT2A and UBQ3 as most significantly enriched hubs in both networks. (**B**) Comparison of “degree –S” for hubs from the interaction networks of genes regulated by –S in WT and OX7.5. “Degree –S” of a hub denotes a number of its interactors with the expression changed by –S. The hubs with a difference between “degree –S” in these two networks amounting to at least 9 and at least 20% in relation to each other are marked by colors. Blue dots indicate hubs with a higher degree –S in OX7.5 than in WT; while green dots mark hubs with a lower degree –S in OX7.5 than in WT. These transcriptomic hubs are described in Appendix A.

**Figure 5 cells-09-00669-f005:**
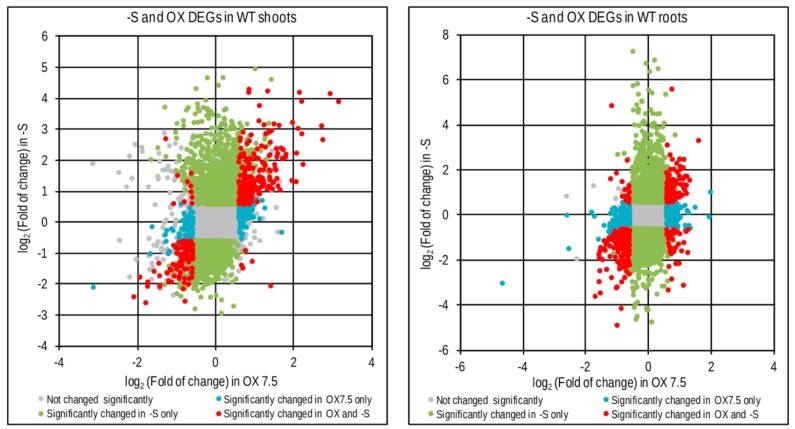
Comparison of genes regulated by –S (-S DEGs) and genes regulated by NBR1 overexpression (OX DEGs) in shoots and roots. Scatterplots show fold of change of individual genes and the effect of sulfur deficit (-S) and NBR1 overexpression (OX) on their expression change.

**Figure 6 cells-09-00669-f006:**
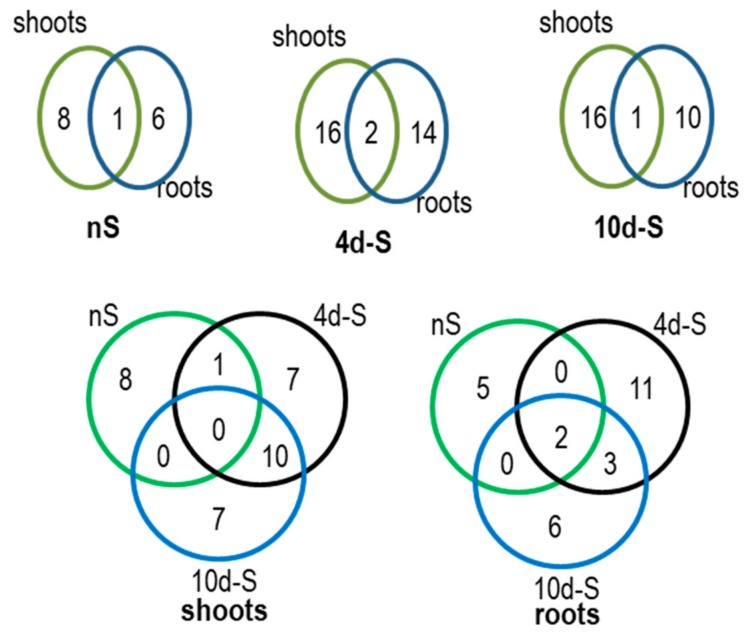
The results of TAP-MS experiment. Venn diagrams indicate the number of proteins co-purifying with NBR1-TAP in the set of 6 interactomes obtained in the TAP-MS experiment. Lists of the proteins (AGI symbols) in all interactomes are shown in Appendix A.

**Figure 7 cells-09-00669-f007:**
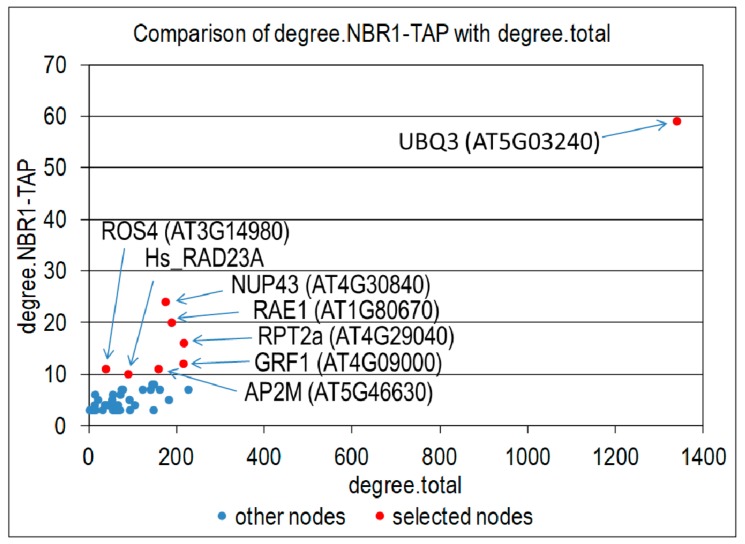
Identification of proteomic hubs. Number of proteins co-purifying with NBR1-TAP (degree. NBR1-TAP) is plotted against the total number of proteins having known interactions with a given network node (degree.total). All network nodes with degree. NBR1-TAP equal 3 or higher are shown as blue dots. All nodes with degree. NBR1-TAP equal 10 or higher were selected as most significant and are shown as red dots, indicated by arrows and labeled.

**Figure 8 cells-09-00669-f008:**
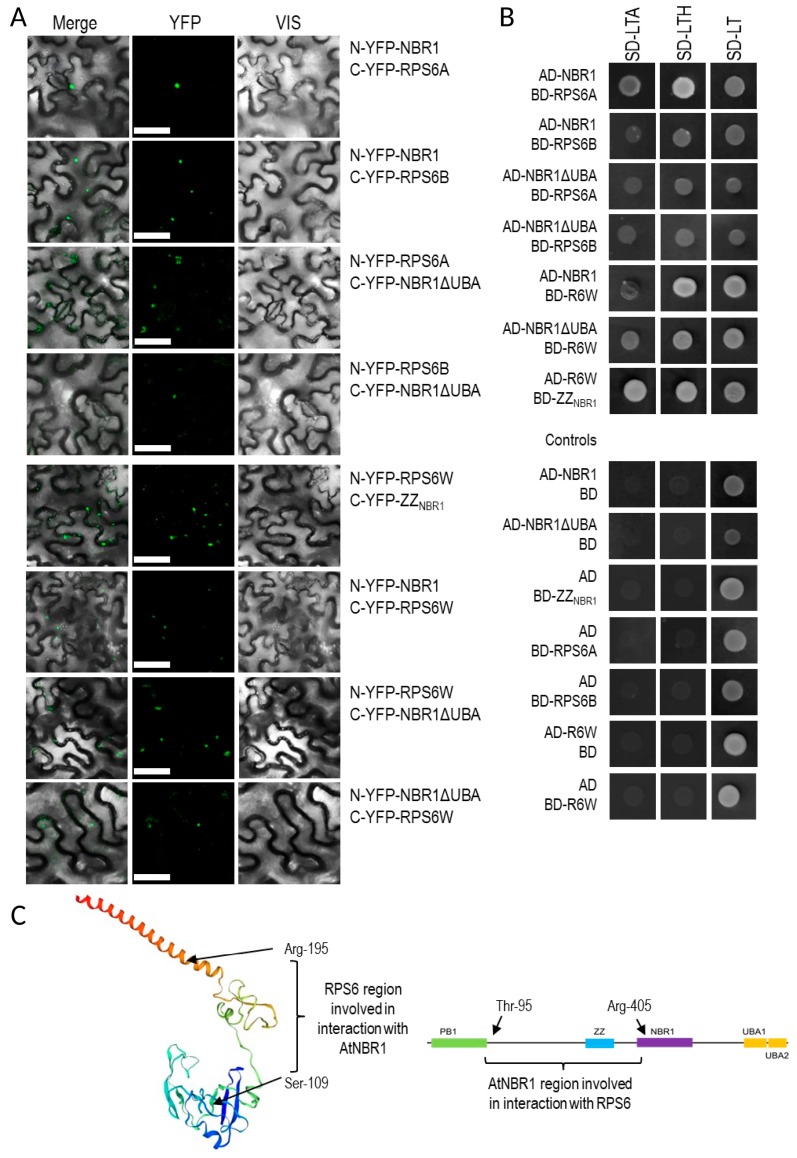
Interaction of NBR1 with RPS6. (**A**) Results of BiFC experiment (the scale bar, 50 µm). The controls are shown in Appendix A. (**B**) Result of Y2H experiment. Full-length NBR1 and NBR1 with C-terminal deletion of 87 amino acids (NBR1ΔUBA), fragments of NBR1 (ZZNBR1) and RPS6B (RPS6W) proteins. (**C**) Scheme showing the protein fragments involved in the interactions. The RPS6 structure is shown as a model of RS62_ARATH protein from SWISS-MODEL. (**D**) BiFC experiment demonstrating intracellular localization of RPS6-NBR1 interactions. RPS6A-NBR1 interaction co-localize in some cases with H2B-RFP, used as a nuclear marker. No colocalization with H2B-RFP was not observed in the case of the RPS6B-NBR1 interaction. The co-localized with H2B-RFP BiFC signals are indicated by arrows, not-colocalized by arrowheads.

**Figure 9 cells-09-00669-f009:**
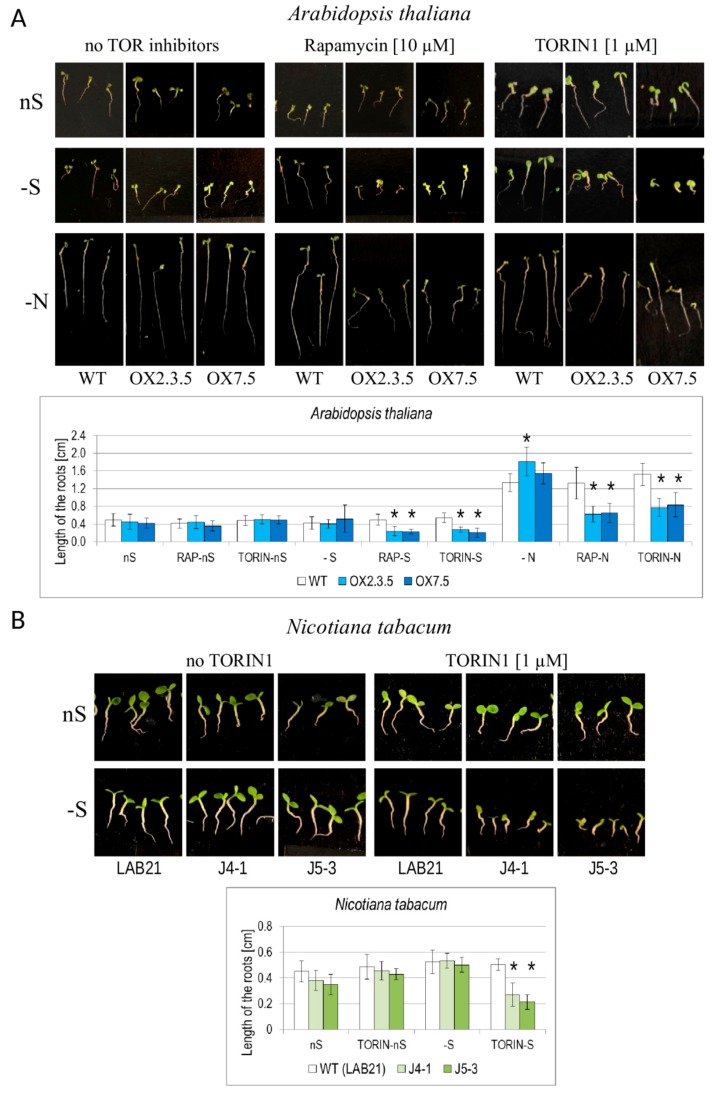
Influence of nutrient starvation and TOR inhibitors on the root lengths of the WT and NBR1-overexpressing lines in *Arabidopsis thaliana* (**A**) and *Nicotiana tabacum* (**B**). The results are from either 6 (rapamycin) or 4 (Torin 1) independednt experiments (6–9 seedlings per experiment).

**Table 1 cells-09-00669-t001:** Overrepresented categories of Gene Family and AraCyc Pathway in the sets of–S DEGs (*p* < 0.01).

	WT-Specific –S DEGs	OX7.5-Specific –S DEGs	–S DEGs from Both Lines
Shoots UP	(*1011 genes*) No significant term	(*459 genes*) No significant term	(*1081 genes*) Receptor kinase-like protein family PP2C-type phosphatase Group A PP2C WRKY Transcription Factors
Shoots DN	(*1351 genes*) No significant term	(*378 genes*) No significant term	(*876 genes*) Kinesins Putative tropine reductase family Sulfurtransferase/Rodanase family
			Chlorophyll biosynthesis
Roots UP	(*959 genes*) Cytoplasmic ribosomal protein gene family	(*667 genes*) Cytoplasmic ribosomal protein gene family	(*1657 genes*) Cytoplasmic ribosomal protein gene family Eukaryotic initiation factor gene family WRKY Transcription Factors Glutathione S-transferase family ABC transporters NAC transcription factors
	tRNA charging		
Roots DN	(*774 genes*) No significant term	(*1007 genes*) No significant term	(*1267 genes*) Aquaporin families MIP families Cytochrome P450
			Glucosinolates biosynthesis Secondary metabolites biosynthesis

**Table 2 cells-09-00669-t002:** Selected hubs with the highest number of interactions with NBR1-interacting proteins (hub degree).

Hub No	AGI Code	Hub Degree	Gene Name/Description (from TAIR, Modified)
Hub1	AT5G03240	59	UBQ3/encodes ubiquitin that is attached to proteins destined for degradation. UBQ3 transcript level is modulated by UV-B and light/dark treatments.
Hub2	AT4G30840	24	NUP43/Transducin/WD40 repeat-like superfamily protein
Hub3	AT1G80670	20	RAE1/Encodes a protein with a DWD motif. It can bind to DDB1a in Y2H assays and may be involved in the formation of a CUL4-based E3 ubiquitin ligase
Hub4	AT4G29040	16	RPT2a/Encodes the 26S proteasome subunit. It is required for root meristem maintenance, and it regulates gametogenesis. It is also shown to regulate gene silencing via DNA methylation.
Hub5	AT4G09000	12	GRF1/Encodes a member of 14-3-3 family. The major native forms of 14-3-3s are homo- and hetero-dimers. Serving as adapters, activators, and repressors, they are involved in a vast array of processes such as the response to stress, cell-cycle control, and apoptosis.
Hub6	AT5G46630	11	AP2M/Clathrin adaptor complexes medium subunit family protein, contains Pfam profile: PF00928 adaptor complexes medium subunit family; it is similar to micro-adaptins of clathrin coated vesicle adaptor complexes
Hub7	AT3G14980	11	IDM1/ROS4/Histone H3 acetyltransferase recognizing unmethylated H3K4 through its PHD domain and methylated DNA through its MBD domain; it negatively regulates DNA demethylation
Hub8	RAD23A	11*	*Homo sapiens* RAD23A having interactions with many Arabidopsis proteins [34]

* including NBR1, identified previously as interactor of RAD23A.

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
