# Peer review of "Overexpression of the Selective Autophagy Cargo Receptor NBR1 Modifies Plant Response to Sulfur Deficit"

_cells, 2020, doi:10.3390/cells9030669_

Round 1
Reviewer 1 Report
This interesting study investigates the effects of the overexpression of NBR1 (Neighbour of BRCA1) in Arabidopsis on gene expression in relation with sulphur supply. The authors identified some NBR1 interacting proteins and found an interaction between NBR1 and the ribosomal RPS6 protein. Finally, NBR1 overexpressing plants appear to be more sensitive to TOR inhibitors upon nutrient starvation.
This manuscript is well written on the overall and covers a lot of ground. However some parts of the manuscript appear disconnected with no efforts to link them in the discussion (transcriptome analysis and interaction with RPS6?). The data and results are in general (see comments) well presented and discussed. Although mainly descriptive, this paper is clearly of interest since it suggests that NBR1 has important roles in plants.
Nevertheless, I have a few points that, to my opinion, should be addressed by the authors:
1- The intensity of the NBR1 bands detected by western blot analysis in Fig.1B should be quantified (on repetitions of the western blot) before reaching a conclusion on the fact that S starvation does not affect NBR1 protein level or that it is higher in one of the transformants. 2- I could not find the number of biological replicates which were used for the transcriptome analyses. If the analyses were performed on a single biological sample for each condition or genotypes I think that the conclusions are not fully valid. Indeed, the observed differences could be merely due to biological variations between samples. In supplementary Table S3, the transcriptome data are compared to data extracted from a submitted paper (Tarnowski et al. on the analysis of NBR1 overexpressors). The submitted paper should be made available to estimate the degree of redundancy between the two papers (if any). 3- The identification of interaction hubs in the genes which were found differentially expressed is interesting but the method for obtaining them is not very clear. Do they correspond to hubs extracted from Arabidopsis interatomic data? I had also difficulties for understanding Fig. 4 . What are the “degrees”? 4- TAP experiments were performed in duplicates but from Table S6 it seems that the reproducibility of the TAP-MS analyses is (as one would expect) quite low. Therefore, the analysis of overlap between lists of interactors shown in Fig. 6 should be performed with proteins found in the two duplicates. 5- The hypersensitivity of root growth to TOR inhibitors in NBR1 overexpressors submitted to N- or S starvation is of interest but should be described in more details. First dose-response curves should be shown to be able to quantitatively compare the effects of TOR inhibitors on growth (see Menand and Montané, JEB 2013, for examples). Torin1 and rapamycin are weak inhibitors in plants and these experiments should be reproduced with more potent inhibitors like AZD8055. Finally these results suggest that TOR activity is lower in nutrient starved NBR1 overexpressors. The measurements of TOR activity could be performed if possible and would strengthen this manuscript. These results should be better discussed as they open new perspectives on the interplay between NBR1 and TOR.Minor points:
The reference list ends at reference 35? In line 75 (Materials and Methods), a nbr1 deletion line is mentioned but does not seem to have been used in this study?Author Response
Please see the attachment.

Reviewer 2 Report
Selective autophagy cargo receptor NBR1 is involved in plant response to sulfur deficit.
In this paper, the authors first report that NRB1 is transcriptionally induced upon sulfur starvation. To explore the potential role of NBR1 in these conditions they (1) analyze the transcriptional responses of plants overexpressing NBR1 and (2) reveal the physical interactors of NBR1 in -S conditions. Results from this study suggest that NBR1 could participate in ribosome remodeling upon sulfur starvation.
Overall the paper is well presented, clear and the data mostly support the author’s conclusions. The paper is of interest, opens new leads for future research and should therefore be considered for publication after further consolidation.
Specifically:
1) I find that the title is a bit too conclusive in regard of the content of the paper – factually, the presented work does not show that NBR1 is involved in plant response to sulfur deficit, it only suggests it. NBR1 overexpression is not physiological and the authors did not show that the absence of NBR1 had an effect on sulfur deficit. The title should be modified accordingly.
2) Data listed below need to be consolidated by providing quantification of 3 independent biological replicates:
-Fig. 1B
-Fig. 2A. Fig.2A represents the average number of acidic compartments per what? picture frame? This is not an optimal representation and it should be replaced by average number of puncta per cell or per area (nm2) of plant that has been analyzed. The authors report that this graph shows the average number of AO-stained spots in “three independent pictures”. Were these pictures taken in one experiment or in 3 independent experiments? In any case, 3 pictures are not enough to provide an accurate representation of a biological phenomenon and perform statistical analyses. A total of at least 100 cells or area should be counted in distinct roots of distinct plants among 3 independent experiments.
-Fig. 2B presents the partial colocalization of NRB1 with AO-stained puncta. Quantification of 3 independent experiments should be provided as the number of NBR1 puncta co-localizing with AO-stained dots.
- Fig. 8A represents BIFC experiments showing an interaction between NBR1 and RPS6A and B. The authors should provide (here or in supplemental information) the appropriate controls with empty vectors showing that the BIFC signal is specific to the interaction of NBR1 with the RPS6 protein and not artefactual.
- Fig. 9. The authors should provide more information on the number of independent experiments and the number of roots analyzed for the graph presented in Fig.9A and 9B.
3) Major results
3.1 NBR1 expression is upregulated by sulfur deficit.
It is clear that NBR1 is transcriptionally upregulated upon sulfur deficiency in physiological conditions (WT cells). How do you explain the increase in NBR1 transcripts in the lines overexpressing NBR1? Is the activity of the 35S promoter sensitive to sulfur deficiency as well or could you consider that the transcripts of NRB1 under post-transcriptional regulation?
Could you provide a likely explanation for the large difference in molecular weight of NBR1 compared to its expected weight? Maybe a covalent complex?
3.2 Seedling from -S have more acridine orange-stained red spots
The reviewer appreciates the authors carefulness in the use and conclusion of AO-staining. Indeed, AO-stained puncta may be autophagosomes but they might as well be other acidic compartments within the cell. The fact that NBR1 co-localizes with some AO-stained puncta further support that these are autophagic structures.
From the reviewer’s point of view the paper would highly benefit from additional experiments in that part of the study to strengthen the argument that – sulfur conditions induce autophagy. First, GFP-ATG8 (or another well described ATG marker) expressing plants should be analyzed upon sulfur starvation conditions to provide conclusive evidence of an induction of autophagosome formation in these conditions. Second/alternatively, the identity of AO-stained puncta as autophagosomes could be challenged by counting the AO-stained puncta in conditions where autophagosome formation is inhibited either genetically (in an atg mutant) or pharmacologically (using Wortmanin).
The authors wrote line 479-480 “identification of the proteins targeted by NBR1 might help to understand the role of selective autophagy in –S conditions”. At this point I am not convinced that we can 100% conclude that –S induces a type of selective autophagy.
3.3 Transcriptional effects of NBR1 overexpression.
I do not quite follow the rationale for looking at changes in gene expression upon NBR1 overexpression. Why would there be any transcriptional responses when NBR1 is actually involved in protein degradation? Further, because NBR1 is physiologically upregulated upon sulfur starvation in WT plants, overexpressing the gene may not cause any effect because its level is not rate-limiting at this point. Did the authors consider looking at nbr1 KO lines in the same conditions?
NBR1 regulates the autophagy degradation of protein cargoes. Therefore, NBR1 ovexrepression may result to a decrease in the level of given proteins that are targeted for autophagy degradation in – sulfur conditions. In that context, quantitative proteomic analyses in the various lines +/- sulfur could be more informative on the cargo/metabolism that is controlled by NBR1 in sulfur deficiency than looking at gene expression. Additionally, data from these experiments could be correlated with interaction data presented in the manuscript to pinpoint likely NBR1-mediated processes upon – sulfur conditions. I consider that these experiments fall out of the scope of the current paper but could nevertheless be discussed in the manuscript.
The authors show that NBR1 and sulfur starvation result in changes in expression of genes encoding ribosomal proteins and therefore suggest that selective autophagy and NBR1 may be involve in ribosome remodeling. Could the authors propose an hypothesis on what would be the role or functional relevance for such remodeling?
3.4 Interactome analyses
Additional information concerning NBR1 should be provided in the introduction notably concerning the relationship between NBR1 and ubiquitin, presentation of the UBA domain (what does UBA stands for, what is its function,..), subcellular localization of NBR1: later in the manuscript the authors identify an interaction of NBR1 in the nucleus and one in the cytosol; what is the significance of such differences in subcellular localization, what do we know about it?
I am convinced that RPS6 interacts with NBR1 from the data presented here – the question is: what does this interaction mean physiologically? Does RPS6 interact with NBR1 preferentially upon –S? Is RPS6 degraded by autophagy in – sulfur condition? Why?
The authors write in the discussion that ‘the biological significance of this novel interaction … remains to be determined’ (l.494). I understand that this is out of the scope of the present study but the authors should still propose hypotheses concerning this biological significance.
3.5 Sensitivity to TOR inhibitors
This part of the paper is not very well described both in the presentation of the results and in the discussion which is a bit light on these results, their significance and the impact of such findings.
4) Minor comments:
Line 434-435; the sentence “Perhaps it would be part…phenomenon” is not clear.
Line 453; the authors send reference to a paper that is not yet published (Wawrzynska and Sirko, submitted), please change to “personal communication” or “unpublished data”.
Round 2
Reviewer 1 Report
This revised manuscript investigates the effects of the overexpression of NBR1 (Neighbour of BRCA1) in Arabidopsis on gene expression in relation with sulphur supply. The authors identified some NBR1 interacting proteins and found an interaction between NBR1 and the ribosomal RPS6 protein. Finally, NBR1 overexpressing plants appear to be more sensitive to TOR inhibitors upon nutrient starvation.
The authors generally addressed the points I raised in a satisfactory manner.
Nevertheless several problems remain in this new submission:
- The staining of roots from NBR1 overexpressors (Figure 2) also appears in the provided manuscript that is submitted elsewhere (Figure 6). These results should be removed from one of the submitted manuscript and just cited!
- AZD8055 at the concentration used (1 microM) should have a strong effect on root growth (Figure S3, see Montané and Menand 2013). The authors should address this issue or redo the experiments, maybe the concentration was wrongly calculated? If this observation is true it could be interesting but it is also rather strange that neither Torin nor rapamycin inhibit root growth in the WT controls? Maybe this due to the growth conditions but a control experiment using vertical plates with solid medium should be performed with the inhibitors to show that they have the expected effects.
- The legends of the supplementary Tables are in Polish.
- Line 286: “we calculated the number”
- The reference list is now scrambled with missing references, two references lists and still ends at ref 36!!
The authors should have taken more care in correcting this manuscript.
